# COVID-19 in Pregnancy and Early Childhood (COPE): study protocol for a prospective, multicentre biobank, survey and database cohort study

Ylva Carlsson [1,2] Lina Bergman [1,2] Mehreen Zaigham [3,4]
Karolina Linden [5] Ola Andersson [6] Malin Veje [7,8] Anna Sandström,[9]
Anna-Karin Wikström,[10] Hanna Östling,[11] Helena Fadl [11] Magnus Domellöf [12]
Marie Blomberg [13] Sophia Brismar Wendel [14,15] Ulrika Åden,[16,17]
Verena Sengpiel [1,2] For the COPE study group

YC and LB are joint first authors.

For numbered affiliations see end of article.

**Correspondence to**
Dr Verena Sengpiel;
verena.sengpiel@obgyn.gu.se

## ABSTRACT

**Introduction** There is limited knowledge on how the SARS-CoV-2 affects pregnancy outcomes. Studies investigating the impact of COVID-19 in early pregnancy are scarce and information on long-term follow-up is lacking.

The purpose of this project is to study the impact of COVID-19 on pregnancy outcomes and long-term maternal and child health by: (1) establishing a database and biobank from pregnant women with COVID-19 and presumably non-infected women and their infants and (2) examining how women and their partners experience pregnancy, childbirth and early parenthood in the COVID-19 pandemic.

**Methods and analysis** This is a national, multicentre, prospective cohort study involving 27 Swedish maternity units accounting for over 86 000 deliveries/year. Pregnant women are included when they: (1) test positive for SARS-CoV-2 (COVID-19 group) or (2) are non-infected and seek healthcare at one of their routine antenatal visits (screening group). Blood, as well as other biological samples, are collected at different time points during and after pregnancy. Child health up to 4 years of age and parent experience of pregnancy, delivery, early parenthood, healthcare and society in general will be examined using web-based questionnaires based on validated instruments. Short- and long-term health outcomes will be collected from Swedish health registers and the parents' experiences will be studied by performing qualitative interviews.

**Ethics and dissemination** Confidentiality aspects such as data encryption and storage comply with the General Data Protection Regulation and with ethical committee requirements. This study has been granted national ethical approval by the Swedish Ethical Review Authority (dnr 2020-02189 and amendments 2020-02848, 2020-05016, 2020-06696 and 2021-00870) and national biobank approval by the Biobank Väst (dnr B2000526:970). Results from the project will be published in peer-reviewed journals.

**Trial registration number** NCT04433364.

## Strengths and limitations of this study

► The COVID-19 in Pregnancy and Early Childhood (COPE) study is a unique linkage between the Swedish Pregnancy Register, the Swedish Neonatal Quality Register, the Hospital Integrated Biobank Sweden and patient-reported outcomes through web-based questionnaires enabling both short-term and long-term follow-up of pregnant women, their partners and children during the COVID-19 pandemic.

► Prospective and automated collection of healthcare data in the comprehensive Swedish Pregnancy Register and Swedish Neonatal Quality Register covering more than 98% of all deliveries in Sweden ensures high-quality data.

► Logistics provided by Hospital Integrated Biobank Sweden enable high-quality biological sampling at several time points during pregnancy according to standardised protocols. However, due to resource limitations at the hospitals during the pandemic, some women will not have complete samples from all time points of interest.

► Based on validated instruments, child health and development during the first 4 years of life will be reported by parents along with comprehensive register-based long-term follow-up. There is a risk of selection bias regarding the follow-up questionnaires where we expect that a proportion of the study population will not answer the questionnaires.

► Other limitations include self-selection bias as women need to give consent to participate, and a prerequisite for participation in the interview part of the study is adequate Swedish or English language skills.

## INTRODUCTION

The emergence of a new coronavirus was brought to WHO's attention on 31 December 2019. Within weeks, a global health emergency ensued and COVID-19 was declared

a pandemic by the WHO. There was an urgent need to identify and protect vulnerable populations within the society and from the knowledge gained from the previous human coronavirus outbreaks of SARS-CoV-1 and Middle East respiratory syndrome CoV, it was clear that pregnant women and their fetuses may be particularly at risk for poor outcomes.[1 2]

## COVID-19 in pregnancy

Recent reviews have found that pregnant women are more likely to need intensive care treatment related to COVID-19 as compared with non-pregnant women.[3 4] Increased severity of disease in late pregnancy along with rapid recovery after delivery has also been reported.[5–7] Pregnant women with COVID-19 are often treated with low-molecular heparin due to a perceived increased risk of thromboembolic events but there is limited evidence.[8] COVID-19 has been found to be associated with a higher prevalence of pre-eclampsia and preterm delivery.[4 9 10] In addition, characteristics like advanced maternal age and high body mass index have been associated with an increased frequency of severe disease[4] and there is insufficient knowledge on fetal malformations and miscarriages related to COVID-19 during early pregnancy. Current guidelines on how to monitor and treat pregnant women and their newborns are based on inadequate evidence with little data from infection in early or mid pregnancy.

## COVID-19 and the offspring

Transplacental transmission of SARS-CoV-2 remains a topic of much debate[11] with several reports suggesting its possibility.[12] Vivanti *et al*[13] and Zaigham *et al*[14] have reported convincing cases of vertical transmission but data on neonatal morbidity and neurodevelopment after SARS-CoV-2 infection during pregnancy are lacking. Although the majority of neonates born to SARS-CoV-2 positive mothers have reported mild, if any, symptoms, several studies have presented a spectrum of clinical symptoms, from mild to severe, in both SARS-CoV-2 positive and negative neonates.[15–18] SARS-CoV-2 is a possible neurotropic virus, and it is well known that congenital or early neonatal infections with neurotropic viruses can impaire brain development.[19] Further, infections in general during pregnancy and in the neonatal period are known to be associated with adverse consequences on brain development and neuropsychiatric diseases.[20 21]

## COVID-19 and childbirth/early parenthood experience

COVID-19 during pregnancy can have a profound impact on how a woman and her partner experience pregnancy, childbirth and early parenthood. The many unknowns connected to the virus have the potential to create anxiety in the pregnant population.[22] International studies have indicated a possible rise in depression among pregnant and lactating women.[23] There have been drastic changes in antenatal and delivery care routines in order to prevent the risk and spread of infection. These changes can have a profound effect on parents and newborns. Examples include limitations in allowing only the parent who gave birth to stay with the newborn at the postnatal ward, or that a newborn in need of neonatal care may be separated from the parents completely until they recover, resulting in the parents missing out on the important first days of bonding with their newborn.[24]

Similarly, severe maternal morbidity may aggravate emotional distress[25] and may be linked to a higher risk of post-traumatic stress disorder in both the mother[26] and the partner.[27] This, in turn, may negatively influence the parent–infant bond and affect subsequent child development.

## Research premises in Sweden and rationale for the study

Antenatal care is offered free of cost for all women in Sweden and follows standardised guidelines.[28 29] Research in Sweden offers the unique possibility to link data from national mandatory health registers, quality registers and registers held by the National Board of Health and Welfare with analyses on biological samples stored in Hospital Integrated Biobanks (SIB, http://www.biobanksverige.se/). Data retrieval and sampling by SIB and pre-established standardised registers with automatic transfer of data from medical records will not impact the workload of the ordinary hospital staff, and therefore, allow the study to proceed during the pandemic where there are already great constraints and limitations on healthcare resources. With almost universal smartphone usage in Sweden, patient-reported outcomes can be safely and efficiently collected using electronically distributed questionnaires.

In summary, uniformity in laboratory testing, use of hospital integrated biobanks with standardised protocols, follow-up of child health and parent-reported outcomes through survey data and linkage of data from national registers will improve knowledge on how COVID-19 can affect the mother, partner, fetus and child. Sweden is one of the few countries with preconditions that can enable an almost population-based follow-up on parent and child health despite the considerable strains imposed by the COVID-19 pandemic.

## AIMS AND OBJECTIVES

The overall aim of this project is to study the impact of SARS-CoV-2 infection on maternal, fetal and child health, as well as experience of pregnancy and parenthood during the COVID-19 pandemic by:

1. Establishing a biobank with biosamples from both pregnant women with COVID-19 and presumably non-infected pregnant women and their infants.
2. Establishing a database of survey-based data linked to Swedish quality and healthcare registers and information from electronic charts for both mother and child.
3. Performing serological, viral and immunological analyses on biobank samples and linking these to maternal, fetal and child outcomes in order to assess short-term and long-term maternal and child health.

4. Collecting prospective data on how women and their partners experience pregnancy, childbirth and early parenthood in the COVID-19 pandemic using validated questionnaires and qualitative interviews.

## METHODS AND ANALYSIS
### Study design and population
The COVID-19 in Pregnancy and Early Childhood (COPE) study is an ongoing Swedish multicentre study, facilitated by the Swedish network for national clinical studies in Obstetrics and Gynaecology (SNAKS, https://www.snaks.se/). Data are collected in four different ways: (1) biosampling, (2) survey-based follow-up until 4 years after delivery, (3) linkage to Swedish health and quality registers enabling long-term follow-up and (4) interviews. All Swedish maternity units with their corresponding neonatal care units have been invited to participate in the study. Centres can participate in the biobank and/or the questionnaire part of the study. So far, 27 maternity units corresponding to more than 87 000 of the approximately 114 000 deliveries per year in Sweden are participating in the study (online supplemental material 1).

Patient recruitment formally started on 1 June 2020. All women, aged 18 years or older, receiving antenatal care or giving birth at participating centres are eligible for the study. Study information and questionnaires have been translated into the most commonly spoken languages in Sweden (Swedish, English, Arabic and Somali). A prerequisite for participation in the interview part of the study is adequate Swedish or English language skills.

Participants have access to study information which is freely available in the waiting rooms of the antenatal care and maternity units involved in the study, the COPE study homepage (www.copestudien.se), social media, interviews and articles available in mass media along with active recruitment by the local study research team.

Recruitment of pregnant women may occur at different time points during the pregnancy, for example, during the first or second trimester ultrasound screening visit, on admission for pregnancy complications or admission to the delivery/COVID-19 units of any of the participating hospitals.

Partners aged 18 years or older are also eligible for participation. Participating women and their partners receive oral and written information about the study and are required to provide written consent. Women can choose to participate in either the biobank or the questionnaire part of the study, or both.

The study is recruiting two groups of women and their partners: (1) a 'COVID-19 group' (figure 1A) and (2) a 'screening group' (figure 1B). The primary goal is to recruit 200 women in the 'COVID-19 group' and 1000 women in the 'screening group'. Further recruitment to the biobank part will depend on adequate funding, study centre capacity and the overall progress of the pandemic. Inclusion to the questionnaire part of the study will continue until the obstetric and neonatal

departments return to their prepandemic routines and social restrictions due to the COVID-19 pandemic are revoked.

### The COVID-19 group
This group includes women that (1) test-positive for SARS-CoV-2 during pregnancy or at delivery, (2) have a positive SARS-CoV-2 antibody test from infection during the current pregnancy or (3) have COVID-19 as a 'clinical diagnose' at the time point of delivery before test results are available.

Before June 2020, there was limited testing capacity in Sweden and only symptomatic patients admitted to the hospital were tested. Since June–July 2020, testing for SARS-CoV-2 has become widely available, even outside hospitals, to all citizens in Sweden. In the beginning of 2021, SARS-CoV-2 screening was introduced for all patients admitted to Swedish hospitals including pregnant women on admission to maternity units. All adults, including pregnant women, are currently required to take a SARS-CoV-2 test in case of symptoms. Detailed data on the number of performed tests and test-positivity in different regions, age groups and over time are available at the homepage of the Public Health Agency of Sweden.[30] Due to restrictions on research-related appointments during the pandemic, women are recruited to the COVID-19 group when they seek inpatient care or in connection with their routine antenatal care visit. In the later case, they are included when they are in remission from COVID-19.

### The screening group
This group consists of women without symptoms and/or with a negative test for SARS-CoV-2 during the current pregnancy. These women are recruited at participating centres during their antenatal care check-up or during their visit to the maternity unit.

A woman in the screening group may be included into the COVID-19 group later on during the pregnancy if she contracts COVID-19. This may also be the case at the time of statistical analyses, in case the biobank specimens should indicate an asymptomatic SARS-CoV-2 infection or a positive test result is found registered in the Swedish Register for mandatory registration of notifiable infectious diseases (SmiNet).

### Biological samples from women and newborns
Table 1 describes the maternal and newborn biological samples that are collected prospectively within the COVID-19 in Pregnancy and Early childhood biobank (COPE biobank). Samples are either sent to local hospital laboratories or to the hospital's biobank facility. Blood samples, liquor samples, amniotic fluid and urine are spun and aliquoted into 225 µL wells. Swabs are frozen in primary tubes and breast milk is vortexed and aliquoted into 0.5 mL wells. Samples are frozen at −80°C within 6 hours. For samples obtained off-hours, these are spun or vortexed, redistributed into secondary tubes and

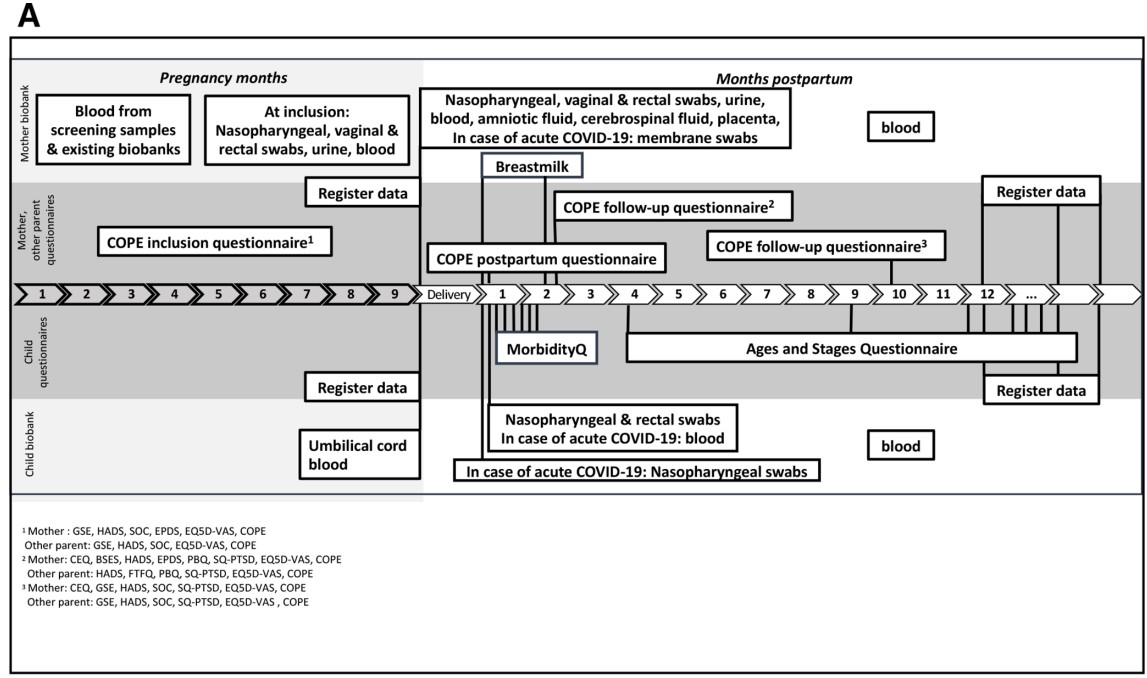

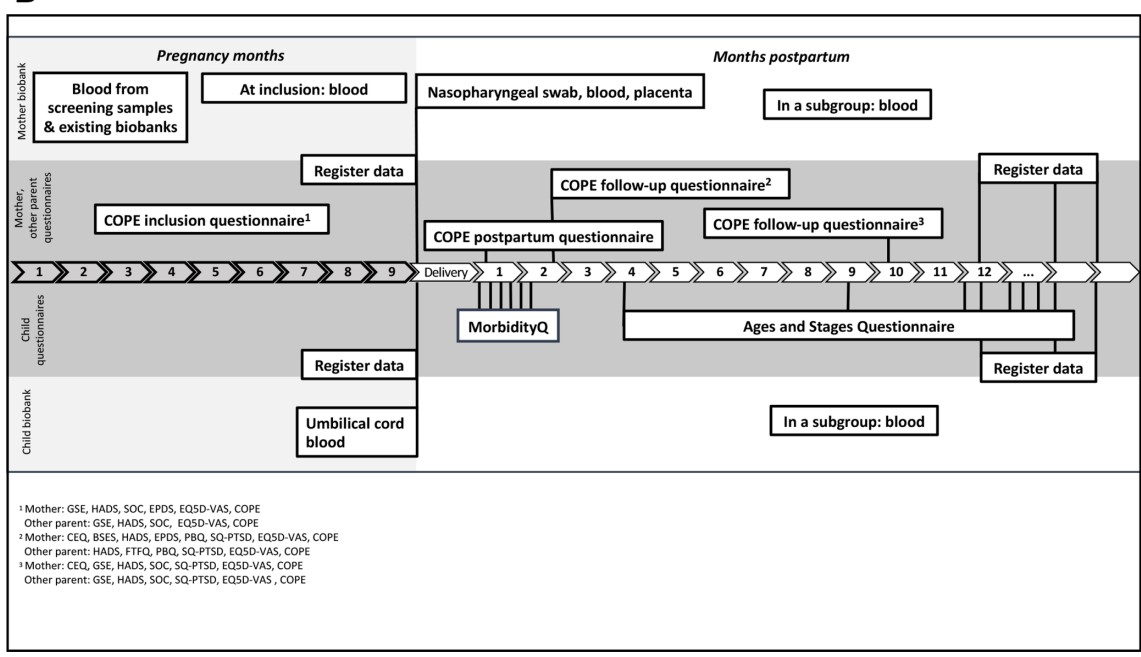

**Figure 1** (A) COVID-19 group: data and biospecimen collection overview. (B) Screening group: data and biospecimen collection overview. BSES, Breastfeeding Self-Efficacy Scale short form; CEQ, Childbirth Experience Questionnaire; COPE, COVID-19 in Pregnancy and Early Childhood; EPDS, Edinburgh Postnatal Depression Scale; EQ5D, EuroQol 5D; FTFQ, First Time Fathers Questionnaire; GSE, General Self-Efficacy scale; HADS, Hospital Anxiety and Depression Scale; SOC, Sense of Coherence Scale; SQ-PTSD, Screen Questionnaire-Post-Traumatic Stress Disorder; VAS, Visual Analogue Scale.

stored in a refrigerator overnight but frozen within 24 hours at the latest. Samples are only thawed directly prior to analysis.

Samples will be analysed with real-time PCR (RT-PCR) for SARS-CoV-2, serology for SARS-CoV-2 along with other immunological analyses that will be specified according to up-to-date techniques pertinent to SARS-CoV-2 infection.

Pre-existing samples taken as part of routine antenatal care and screening will be obtained from already-existing biobanks along with the newly established COPE biobank. As part of routine antenatal care in Sweden, all women provide a blood sample that is used to screen for hepatitis B, syphilis, rubella and HIV infection in early pregnancy. These samples are stored in Hospital biobanks, according

**Table 1** Maternal and newborn biological samples in the COVID-19 in Pregnancy and Early childhood biobank (COPE biobank)

| Time point | Sample |
| --- | --- |
| **COVID-19 group—mothers** | |
| Antenatal screening | Pre-existing samples taken as part of routine antenatal care and screening. More details under 'biological samples from women and newborns'. |
| On being diagnosed with SARS-CoV-2 infection | Nasopharyngeal and pharyngeal swabs or saliva |
| | Blood 30 mL |
| | Vaginal swab |
| | Rectal swab |
| | Urine 10 mL |
| At delivery or in case of pregnancy loss/termination of pregnancy | Nasopharyngeal and pharyngeal swabs or saliva |
| | Blood 30 mL |
| | Placenta, 12 pieces, in total approximately 10–15 cm$^3$ |
| | Vaginal swab |
| | Rectal swab |
| | Urine 10 mL |
| | Placenta/membrane swab (in case of COVID-19 within 14 days before delivery or COVID-19 diagnosis up to 2 days after delivery) |
| In case of Caesarean section | Amniotic fluid 10 mL |
| In case of spinal anaesthesia, e.g. at caesarean section | Cerebrospinal fluid 5 mL |
| At 48–96 hours follow-up post partum | Breast milk 5–10 mL |
| Follow-up within 12 months post partum | Blood 10 mL |
| | Breast milk 5–10 mL |
| **COVID-19 group—children** | |
| At birth | Umbilical cord blood 7 mL In case of stillbirth: Nasopharyngeal and pharyngeal swabs or saliva and blood 5 mL from heart puncture. These samples are routinely performed as standardised clinical practices in case of stillbirth. |
| Within 12 hours of delivery | Nasopharyngeal and pharyngeal swabs (in case of maternal COVID-19 within 14 days before delivery or COVID-19 diagnosis up to 2 days after delivery) |
| 48–96 hours post partum | Nasopharyngeal and pharyngeal swabs |
| | Rectal swab |
| | Blood 5 mL (in case of maternal COVID-19 within 14 days before delivery or COVID-19 diagnosis up to 2 days after delivery) |
| Follow-up within 12 months post partum | Blood 5 mL |
| **Screening group—mothers** | |
| Antenatal screening | Pre-existing samples taken as part of routine antenatal care and screening. More details under 'biological samples from women and newborns'. |
| Follow-up antenatal screening | Blood 30 mL |
| Delivery | Blood 30 mL |
| | Nasopharyngeal and pharyngeal swabs or saliva |
| | Placenta in a subgroup of women as controls, 12 pieces, in total approximately 10–15 cm$^3$ |
| A subgroup of women in the screening group (n=30) are sampled according to the COVID-19 group protocol as 'controls' | |
| **Screening group—children** | |
| At birth | Umbilical cord blood 5 mL (plasma) and 1.5 mL (cells). In case of stillbirth: nasopharyngeal and pharyngeal swabs or saliva and blood 5 mL from heart puncture. These samples are routinely performed as standardised clinical practices in case of stillbirth. |
| A subgroup of newborns in the screening group (n=30) are sampled according to the COVID-19 group protocol as 'controls' | |

COPE, COVID-19 in Pregnancy and Early Childhood.

to the Swedish Biobanks Medical Care Act (https://biobanksverige.se/english/research/). For women and children included in the study, these samples may be analysed for serology for SARS-CoV-2 in order to define the time point for an asymptomatic SARS-CoV-2 infection. Similarly, in some centres in Sweden, blood samples from routine testing for immunisation at gestational week 28 are stored and may be analysed for serology for SARS-CoV-2. Further, we plan to combine already existing pregnancy biobanks that have biosamples from different gestational weeks (IMPACT study, dnr 2018-231 (http://www.impactstudien.se/), Uppsala/Örebro; GO PROVE, dnr 955-18, Gothenburg; UPMOST, dnr 2019-00309, Uppsala/Örebro and Biobank för gravida kvinnor, Uppsala, dnr 2007-181, Uppsala/Örebro) with the COPE biobank. These already existing biobanks will provide blood samples collected from February 2020 onwards.

### Register and medical record data on obstetric, medical and neonatal outcomes

Biobank laboratory analyses and questionnaire results will be linked to register data using Swedish personal identification numbers in order to follow long-term maternal and child health as well as child growth and development.

Data will be linked to national quality registers, for example, data on pregnancy outcome, neonatal health or maternal intensive care unit admission (Swedish Pregnancy Register (SPR), Neonatal Quality Register (SNQ) and Intensive Care register), registers from the National Board of Health and Welfare, for example, data on long-term child health (National Patient Register, National Cause of Death Register, Prescribed Drug Register), Statistics Sweden, for example, data on education and income (Register of Total Population, Education Register and Income Register, LISA register), Public Health Agency, data on time point for testing positive for SARS-CoV-2 (Swedish Register for obligatory registration of notifiable infectious disease; SMiNet), growth data during childhood (the Swedish Child Health Care Register) and medical records for additional data, for example, cardiotocography during delivery. The registers are described in detail in box 1.

### Questionnaires

On inclusion, women and their partners from both the COVID-19 group and the screening group, are asked to fill out different electronic questionnaires up to 4 years after delivery. Questionnaires include questions specifically developed for the COPE study as well as validated surveys in order to test for differences between the two groups. Questions developed specifically for COPE have been translated into English, Arabic and Somali. Questions based on validated surveys are provided in other languages only if there is a validated version available.

### Patient-reported outcome measures

Participants are asked about COVID-19 related symptoms, COVID-19 in household members, their work situation, physical activity, the general impact COVID-19

---

**Box 1   Swedish national health and quality registers that will be linked to the COVID-19 in Pregnancy and Early Childhood dataset (COPE dataset)**

**The Swedish Pregnancy Register**
A certified national quality register initiated by the Swedish healthcare regions that combines prospectively collected data from the Swedish Maternal Health Care Register, the Swedish National Quality Register for Prenatal Diagnosis and data from electronic, standardised, prenatal delivery and neonatal records. The register includes more than 95% of all deliveries in Sweden and covers pregnancies from the first antenatal care visit until the follow-up visit at 8–12 weeks post partum. It contains information on maternal characteristics, medical and reproductive history, pregnancy examinations, delivery outcomes and follow-up[41] (http://www.graviditetsregistret.se/).
Examples of variables that will be extracted: pregnancy loss after first visit to antenatal care, body mass index at booking visit, weight gain during pregnancy, gestational age at delivery, mode of delivery, postpartum blood loss, birth weight, Apgar score and pregnancy complications such as gestational hypertension, pre-eclampsia and gestational diabetes.

**The Swedish Neonatal Quality Register**
A national quality register for neonatal care. All neonatal departments in Sweden report standardised data on admitted infants including basic information about pregnancy and childbirth, as well as the condition, treatment and diagnoses of the infant according to the Swedish version of International Classification of Diseases 10th revision (ICD-10), as well as information from follow-up visits. During the COVID-19 pandemic, all children born to mothers testing positive for SARS-CoV-2 are registered in the Swedish Neonatal Quality Register[42] (www.snq.se).
Examples of variables that will be extracted: admission to neonatal intensive care unit (NICU) or neonatal special care (NSC), duration of NICU/NSC stay, mechanical ventilation, asphyxia related complications, hypothermia treatment.

**The Swedish Intensive Care Register**
A Swedish quality register on intensive care. Data regarding severity of disease and interventions will be retrieved from Swedish Intensive Care Register for women requiring care at an ICU (http://www.icuregswe.org/).
Examples of variables that will be obtained: duration of stay, mechanical ventilation, extracorporeal membrane oxygenation.

**The National Patient Register**
A mandatory health register including diagnoses on hospital admissions and outpatient visits in specialist care. Information will be retrieved on ICD-10 diagnoses and interventions for women during pregnancy and the postpartum period, chronic or previous disease in the mother as well long-term follow-up of their children[43] (https://www.socialstyrelsen.se/statistik-och-data/register/alla-register/patientregistret/).
Examples of variables that will be extracted: ICD-10 diagnosis of COVID-19 or diagnosis of thromboembolism for the mother during pregnancy as well as 3 months postpartum, diagnosis of neuropsychiatric disease during childhood.

**The National Cause of Death Register**
The national cause of death register is based on the obligatory death certificates that need to be signed by a medical doctor confirming the cause of death in the deceased. Both the date and cause of death are registered.

Continued

---

## Box 1  Continued

Examples of variables that will be extracted: Time point and cause of death for mothers in the study population until 42 days after delivery, time point and cause of neonatal deaths.

### The Swedish Prescribed Drug Register
A mandatory register holding data on all prescribed substances, Anatomical Therapeutic Chemical classification-code and date of purchase, for all dispensed drugs in the outpatient population.[44]
Examples of variables that will be extracted: antibiotics prescribed to children during the first year of life.

### The Swedish Register for mandatory registration of notifiable infectious disease (SmiNet)
COVID-19 is classified as a notifiable infectious disease. All positive SARS-CoV-2 tests are reported to SmiNet by the laboratories analysing the tests as well as the medical doctor responsible for sampling (https://www.folkhalsomyndigheten.se/smittskydd-beredskap/overvakning-och-rapportering/sminet/).
Variables that will be extracted: Date for positive SARS-CoV-2 test or date of COVID-19 disease during pregnancy.

### The Swedish Child Health Care Register
A national quality register on child healthcare (http://bhvq.se/).
Variables that will be extracted: length and weight at 3, 4 and 5 years of age.

### Statistics Sweden
Statistics Sweden is a government agency collecting data on education and income (https://www.scb.se/en/).
Variables that will be extracted: family income, education level of the study participants, child school grades.

has had on their lives and how they experience social isolation. Further, validated questions from the Gothenburg Research Programme on pregnancy and politics concerning study participants' opinion on the healthcare sector and authorities during the pandemic[31] and free-text questions are also asked. Free-text answers will be analysed using content analysis methodology.[32] Based on validated questionnaires, the study participants are asked to rate their self-efficacy, health-related quality of life, sense of coherence, anxiety/depression, childbirth experiences, levels of breastfeeding self-efficacy, parent–infant bonding, symptoms of post-traumatic stress, self-esteem, perceived stress and attachment style. For details, see table 2.

### Parent-reported infant morbidity and development
A weekly, web-based child morbidity questionnaire is sent out to parents during the first 6 weeks after delivery. Symptoms of infection (fever 38.0°C or more), abdominal, airway and other symptoms (otitis, rash, excessive crying, tiredness), visiting a doctor, prescription of antibiotics or whether the child has been admitted to hospital are noted. The questionnaire has previously been used in two other Swedish studies.[33 34]

At 4, 9 and 12 months (corrected age) along with 2, 3 and 4 years of age, parents report their infant's development based on the validated Ages and Stages Questionnaire (ASQ)-version III[34–36] (see table 2).

### Interviews: women's and their partners' experiences of pregnancy, childbirth and postnatal care during the COVID-19 pandemic
Informants will be approximately 12–20 women who were diagnosed with COVID-19 during pregnancy and their partners, as well as 12–20 women and partners who were not diagnosed with COVID-19. Participants will be selected to ensure a broad range of views and experiences, for example, age, parity and socioeconomic background including severity of symptoms in the COVID-19 group. The women and their partners will be interviewed separately using face-to face interviews or by video link or telephone. Open-ended questions with follow-ups will be asked to deepen the understanding.[32] Interviews will last approximately 1 hour, and will be audiotaped and transcribed verbatim. Data analysis will be conducted using either phenomenology with a lifeworld approach[37] or by using content analysis.[32]

## DATA PROCESSING AND ANALYSIS
### Research questions
The COPE study is collecting information for a database and biobank in order to study the association of COVID-19 during the pregnancy with a wide variety of different pregnancy, maternal and neonatal outcomes including the long-term follow-up of maternal and child health, as well as parental experience.

Predefined research questions concern the incidence of infection and COVID-19 at different time points of the pandemic; the impact of COVID-19, gestational age at infection, severity of disease, viral load, presence of SARS-CoV-2 and/or antibodies against SARS-CoV-2 in different compartments of the mother and/or child, pregnancy outcome, maternal and child health; experience of childbirth and early parenthood during the pandemic (see online supplemental material 2 for details).

### Exposure definition in regard to clinical outcomes
Based on RT-PCR and serology and data from SmiNet, women will be divided into infected women (COVID-19 during pregnancy) and non-infected women (no COVID-19 during pregnancy). In some analyses/subanalyses, gestational age at infection, severity of disease, viral load and immune response will be considered as additional exposure variables for the infected group.

### Examples of outcome definition
▶ Pregnancy and neonatal outcomes will be retrieved from the SPR and SNQ, either registered as tick boxes, actual measures or International Classification of Diseases 10th revision (ICD-10) codes: Pre-eclampsia (ICD-10 O14), gestational age at birth, preterm

**Table 2** Overview and time frame of data sampling by questionnaires

| Questionnaire time point | Mother | Other parent |
|---|---|---|
| Inclusion (pregnancy), approximately 20 min | GSE*†, HADS*†, SOC*†, EPDS*†, EQ5D-VAS*† | |
| | COPE questionnaire*†‡: Demographic variables, COVID-19 symptoms, physical activity, subjective experiences and free-text option | |
| 1-week post partum, approximately 5 min | COPE questionnaire: hospital stay, COVID-19 symptoms, hygiene measures etc.*†‡ | |
| 1, 2, 3, 4, 5, 6 weeks post partum, approximately 5 min | COPE questionnaire: Morbidity-Q*†‡<br>At 6 weeks also COVID-19 symptoms | |
| 8–12 weeks post partum, approximately 30 min | CEQ*, BSES*, EPDS*† | FTFQ* |
| | HADS*†, PBQ*, SQ-PTSD*, EQ5D-VAS*† | |
| | COPE questionnaire*†‡: demographic variables, COVID-19 symptoms, physical activity, subjective experiences and free-text option | |
| 4 months (corrected age), approximately 10 min | ASQ* | |
| 9 months (corrected age), approximately 10 min | ASQ* | |
| 10 months post partum, approximately 25 min | CEQ* | |
| | GSE*†, HADS*†, SOC*†, SQ-PTSD*, EQ5D-VAS*† | |
| | COPE questionnaire*†‡: demographic variables, physical activity, subjective experiences and free-text option | |
| 12 months (corrected age), approximately 10 min | ASQ* | |
| 2 years post partum, approximately 25 min | ASQ* | |
| | HADS*, RSES*, PSS*, EQ5D-VAS*, ECRS*<br>COPE questionnaire: demographic variables | |
| 3 and 4 years post partum, approximately 25 min | ASQ* | |
| | HADS*, RSES*, PSS*, EQ5D-VAS*<br>COPE questionnaire: demographic variables | |

*Available in English.
†Available in Arabic.
‡Available in Somali.
ASQ, Ages and Stages Questionnaire-Version III; BES, Breastfeeding Self-Efficacy Scale; CEQ, Childbirth Experience Questionnaire; COPE, COVID-19 in Pregnancy and Early Childhood; ECRS, Experiences in Close Relationsships Scale; EPDS, Edinburgh Postnatal Depression Scale; FTFQ, First Time Fathers Questionnaire; GES, General Self-Efficacy scale; HADS, Hospital Anxiety and Depression Scale; PBQ, Postpartum Bonding Questionnaire; PSS, Perceived Stress Scale; RSES, Rosenberg Self-esteem Scale; SOC-13, 13-item Sense of Coherence Scale; SQ-PTSD, Screen Questionnaire - Post-Traumatic Stress Disorder; ; VAS, Visual Analogue Scale.

delivery (with subanalyses for early, moderate, late preterm delivery as well as spontaneous vs iatrogenic preterm delivery), birth weight, small for gestational age, birth asphyxia, and perinatal death.
► Thromboembolic event diagnosis will be retrieved from the SPR and the National Patient Register (ICD-10 I82, I26).
► Vertical transmission as defined by Shah *et al.*[38]
► Child development and health: Developmental delays or potential delays as measured by ASQ up to 4 years of age. Neurological disorders diagnosed during the first 4 years of life (composite) retrieved from the National Patient Register; Any mental or behavioural disorder (ICD-10 F00–F99), impaired vision (H54), impaired hearing (H90, H91), cerebral palsy and other paralytic syndromes (G80–G83).
► Presence of antibodies in umbilical cord blood (IgG, IgM) and breast milk (IgG, IgM, IgA).

Subanalyses will be performed to study the impact of country of birth, socioeconomic status or underlying disease in the mother (eg, obesity, hypertension, diabetes, asthma).

### General statistical methodology

Demographics will be presented as numbers (percentage), medians or means as appropriate by distribution. Comparisons between groups will be analysed by Student's t-test or Mann-Whitney U-test with means or medians and confidence intervals or IQR, as appropriate according to distribution of the variables. Categorical variables will be compared by $\chi^2$ test or Fisher's exact test. Correlations will be analysed by Pearson's r or Spearman's r as appropriate by distribution of the variable. Regression analyses, unadjusted and adjusted, will be performed to adjust for known confounding variables. Ten cases per variable at the lowest will be considered appropriate to avoid overfitting of the model.

**Table 3** Assumptions for power calculation

| Outcome | Prevalence in screening group | Prevalence in COVID-19 group | Sample size screening group | Sample size COVID-19 group |
|---|---|---|---|---|
| Pre-eclampsia | 6% | 12% | 511 | 256 |
| Thrombosis | 0.13% | 2% | 613 | 306 |
| Small for gestational age (below 10th percentile) | 10% | 20% | 199 | 199 |
| Preterm delivery | 6% | 12% | 511 | 256 |
| Neurological and/or neurodevelopmental disorders | 1% | 4% | 581 | 290 |

## Sample size calculations

This is an exploratory study where the initial sample size for recruitment is set to 200 women in the COVID-19 group and 1000 in the screening group (with an additional 10%–15% with assumed positive tests in the screening group thus more than 300 COVID-19 cases). Sample size has been calculated with 80% power and a significance level of <0.05 based on the assumptions shown in table 3. As outcomes are retrieved from population-based registers, there will be close to complete follow-up.

The percentage of women with vertical transmission secondary to active COVID-19 at the time of delivery (14 days before to 2 days after delivery) will be calculated.

Depending on the prevalence of malformations, the presence and type of malformations in case of infection during early pregnancy will be presented in a purely descriptive manner.

The presence of antibodies in umbilical cord blood and breast milk will be studied in relation to gestational week of infection, severity of infection and maternal serum antibody levels. Antibody levels will be followed over time.

Due to the unprecedented nature of the pandemic, it is difficult to perform formal power calculation with regard to parent health and mental well-being.

## PATIENT AND PUBLIC INVOLVEMENT

A pregnant patient representative, her partner and a patient organisation namely, the Swedish Association for Premature Infants (Svenska prematurförbundet) were invited to participate in the early stages of planning the study. Several video meetings were held between a research group representative (KL) and the patient representative and her partner where the research questions and outcomes were discussed. Participant time investment in the study was discussed and adjustments in the questionnaires were made according to their feedback. The opinion and feedback obtained from the non-pregnant parent were also given special importance.

The study website (www.copestudien.se) was designed in collaboration with the patient representative and provides a convenient platform for participants to connect with the research team. Communication with the wider pregnant population and the public through social and mass media has enabled uptake of patient-evoked research questions and led to appropriate modifications in the study protocol.

## ETHICS AND DISSEMINATION

COPE is a comprehensive cohort study involving the use of register based data, biological sampling, questionnaires and interviews. The study has received national ethical approval by the Ethics Review Board, Lund, Sweden (dnr 2020-02189 and amendments 2020-02848, 2020-05016, 2020-06696 and 2021-00870) and national biobank approval at Biobank Väst (dnr B2000526:970). Confidentiality aspects such as data encryption and storage comply with the general data protection regulation. All data are stored in a secure online database provided by MedSciNet (www.medscinet.com), an international company specialising in web applications in the field of academic medicine. Biobank samples will be identified using the personal identification numbers of the patients included in the study and pseudonymised after identification in the biobank before laboratory analyses.

Several blood and tissue samples are being collected in the study. Certain samples are collected as part of the standardised diagnostic protocols of the COVID-19 pandemic (eg, nasopharyngeal swabs) while others are study specific (eg, blood samples from infants up to the age of 12 months). Sampling will be performed by experienced nurses and the application of a topical anaesthetic patch will help minimise pain for the child.

Results from this study will be presented at different national and international conferences, in peer-reviewed journals and in mass media. Wide-spread public interest in COVID-19-related research will help facilitate study dissemination and participation from all major Swedish maternity units.

## DISCUSSION

Knowledge from previous coronavirus outbreaks[1 4] has identified pregnant women as particularly susceptible to negative outcomes. While evidence is increasing on how SARS-CoV-2 infection in pregnancy can affect pregnancy outcomes,[9] there is a need to thoroughly examine the burden of COVID-19 during the pregnancy with focus not only on the pregnant woman and the fetus but also the

child after delivery and the pregnant woman's partner. In addition, long-term consequences of COVID-19 during pregnancy are also relatively unknown.

Local and regional differences in the testing and management of pregnant women with SARS-CoV-2, make the results of published studies difficult to interpret. Many studies do not report the gestational age at infection, do not include a control group, recruit women only when they seek healthcare for obstetric complications or on admission for delivery, do not specify the indication for an obstetric intervention (intervention with or due to COVID-19) or are not generalisable since they are based on a single site with a population having a certain risk profile regarding, for example, socioeconomic status or pregnancy complications.[4] It can, therefore, be argued that current guidelines on how to monitor and treat pregnant women and their neonates are based on inadequate evidence with little or no data on long-term follow-up.

Few studies have focused on women's experience of pregnancy, childbirth and early parenthood during the COVID-19 pandemic and the results are often context specific, making them difficult to generalise. Partner experience has been largely overlooked[22] and there is a lack of data on pregnancy and child outcomes secondary to SARS-CoV-2 infection in early pregnancy. Viral infections have been known to cause developmental problems for the fetus such as deafness (cytomegalovirus)[39] or anaemia (parvovirus).[40] Due to the novelty of SARS-CoV-2, the effect of the virus on fetuses in the first trimester is largely unknown.

However, the study design has certain limitations. Although the majority of all maternity units are participating, the study will not be completely representative of the total population. Since women and partners need to actively consent to participate, there will be self-selection bias. Additionally, the questionnaire and interview parts of the study require adequate language skills as described earlier. During the 4-year follow-up period, a certain degree of 'drop-out' is expected in the questionnaire part of the study. With regard to the biobank part of the study, we expect maternity units to show considerable variation in retrieving a full list of samples as the healthcare sector is under enormous pressure due to the pandemic.

To summarise, the COPE study will help lay the foundations for understanding the society's ability to protect one of its most vulnerable groups, pregnant women and their children.[22] The COPE database and biobank will help answer important questions regarding short and long-term complications secondary to COVID-19 in pregnancy and can be used for future research on the prevention of other pregnancy complications after viral infections. The collaboration and research infrastructure built within the COPE study has the potential to facilitate future research within obstetrics and neonatology in Sweden.

**Author affiliations**
[1]Department of Obstetrics and Gynaecologyhe, Institute of Clinical Science, Sahlgrenska Academy, University of Gothenburg, Gothenburg, Sweden
[2]Department of Obstetrics and Gynaecology, Sahlgrenska University Hospital, Region Västra Götaland, Gothenburg, Sweden
[3]Department of Clinical Sciences Lund, Lund University, Lund, Sweden
[4]Department of obstetrics and gynecology, Skåne University Hospital, Malmö, Sweden
[5]Institute of Health and Care Sciences, Sahlgrenska Academy, University of Gothenburg, Gothenburg, Sweden
[6]Department of Clinical Sciences Lund, Pediatrics, Lund University and Skåne University Hospital, Malmö, Sweden
[7]Department of Infectious Diseases, Institute of Biomedicine, Sahlgrenska Academy, University of Gothenburg, Gothenburg, Sweden
[8]Department of Infectious Diseases, Sahlgrenska University Hospital, Gothenburg, Sweden
[9]Department of Medicine, Solna, Clinical Epidemiology Division, Karolinska Institutet, Stockholm, Sweden
[10]Department of Women's and Childen's Health, Uppsala University, Uppsala, Sweden
[11]Department of Obstetrics and Gynaecology, Faculty of Medicine and Health, Örebro University, Örebro, Sweden
[12]Department of Clinical Sciences, Pediatrics, Umeå University, Umeå, Sweden
[13]Department of Obstetrics and Gynecology and Department of Biomedical and Clinical Sciences, Linköping University, Linköping, Sweden
[14]Department of Clinical Sciences, Karolinska Institutet Danderyd Hospital, Stockholm, Sweden
[15]Department of Women's Health, Danderyd Hospital, Stockholm, Sweden
[16]Department of Women's and Children's Health, Karolinska Institutet, Stockholm, Sweden
[17]Neonatal unit, Karolinska University Hospital, Stockholm, Sweden

**Acknowledgements** We would like to thank all families participating in the COPE study, all staff involved in the recruitment and sampling of patients at the different maternity units, the patient representatives, Biobank Sweden and especially Biobank Väst for support in setting up the study and collection of biological specimens.

**Collaborators** For the COPE study group: Anders Elfvin, MD, Associate Professor (Department of Pediatrics, The Queen Silvia Children's Hospital, Sahlgrenska University Hospital, Gothenburg, Sweden and Department of Pediatrics, Institute of Clinical Sciences, Sahlgrenska Academy, University of Gothenburg), Anna Hagman, MD, PhD (Department of obstetrics and gynaecology, Sahlgrenska Academy, chair for working group for perinatology, Gothenburg), Anna Sand, MD, PhD (Karolinska University Hospital, Stockholm), Anna Wessberg, Midwife, PhD (Department of obstetrics, Sahlgrenska University Hospital, Gothenburg), Anna-Carin Wihlbäck, MD, PhD (Department of Clinical Sciences, Obstetrics and Gynecology, Umeå University) Elin Naurin, PhD (Department of Political Science, University of Gothenburg, Gothenburg), Emelie Ottosson, MD (Department of obstetrics and gynecology, Skaraborgs Hospital, Skövde), Emma Von Wowern, MD, PhD (Senior Consultant OBGYN, Inst. of Clinical Sciences Malmö, Department of Obstetrics and Gynecology, Skåne University Hospital, Lund University, Malmö/Lund), Fredrik Ahlsson (Associate Professor, Senior Lecturer, Senior Consultant in Neonatology, Department of Women's and Children's Health, Section for Pediatrics, Uppsala University Children's Hospital), Gilda Dumitrescu, MD (Department of obstetrics and gynecology, Eskilstuna Hospital), Gustaf Biasoletto, MD, Department of Obstetrics and Gynecology, Östersund Hospital), Jan-ÅÅke Liljeqvist, MD (Associate Professor (Department of Clinical Microbiology, Sahlgrenska University Hospital, Gothenburg), Johan Berg, MD (Department of Clinical Sciences Lund, Pediatrics/Neonatology, Lund University and Department of Neonatology, Skåne University Hospital, Malmö/Lund), Johanna Berg, MD (Department of Obstetrics and Gynecology, Norra Älvsborgs Hospital, Trollhättan), Kathrin Rothbarth, MD (Department of Obstetrics and Gynecology, Norra Älvsborgs Hospital, Trollhättan), Kristina Pettersson, MD (Karolinska institutet, Clintec, department of obstetrics and gynecology; Karolinska University hospital, women's health, Stockholm), Linda Hjertberg, MD (Department of Obstetrics and Gynecology, Vrinnevi Hospital, Norrköping, Sweden and Department of Biomedical and Clinical Sciences, Linköping University, Linköping), Linda Iorizzo, MD (Institute of Clinical Sciences Lund,Lund University, Department of obstetrics and gynecology, Helsingborg Hospital, Region Skane), Magnus Lindh, MD (Professor, Department of Clinical Microbiology, Sahlgrenska University Hospital, Gothenburg), Maria Jonsson, MD (Associate Professor, Department of Women's and Children's Health, Uppsala University, Uppsala), Maria Lindqvist, RNM,

PhD (Department of Nursing and Department of clinical sciences, obstetrics and gynecology, Umeå university, Umeå), Maria Revelj, MD (Department of obstetrics and gynaecology Sahlgrenska University Hospital, Gothenburg), Maria Svenvik, MD (Department of obstetrics and gynaecology Kalmar County Hospital and Linköping University), Marie Vikström Bolin, MD, PhD (Department of obstetrics and gynaecology, Sundsvalls hospital), Marie-Charlotte Nilsson, Midwife (Department of Obstetrics & Gynecology, Ystad hospital), Matilda Friman Mathiasson, MD (Department of Obstetrics & Gynecology, Kristianstad hospital), Merit Kullinger, MD (Department of obstetrics and gynaecology Västerås, Västmanlands hospital), Mirjana Janes-Nakic, MD (Neonatologist, Department of Pediatrics and Neonatology, South Älvsborg Hospital, Borås), Ove Karlsson, MD, PhD (NU Hospital Group, Trollhättan and Department of Anesthesiology and Intensive Care, Institution of Clinical Sciences, Sahlgrenska Academy, University of Gothenburg), Pihla Kuusela, MD, PhD (South Älvsborg Hospital, Department of Gynecology and Obstetrics), Sandra Holmström, MD (Department of obstetrics and gynaecology Halland's hospital Varberg), Sofie Graner, MD, PhD (Centre for Pharmacoepidemiology, Department of Medicine Karolinska Institute Solna Stockholm), Susanne Woxenius, MD, PhD (Region Västra Götaland, Sahlgrenska University Hospital, Department of Infectious Diseases, Gothenburg and Department of Infectious Diseases, Institute of Biomedicine, Sahlgrenska Academy, University of Gothenburg, Gothenburg), Thomas Abrahamsson, MD (Associate Professor, Neonatologist, Associate professor, Department of Pediatrics and Department of Biomedical and Clinical Sciences, Linköping University), Åsa Leijonhufvud, MD, PhD (Department of Clinical Sciences Lund/Clinical Science Helsingborg. Helsingborgs Lasaret), Åsa Pontén, MD (Department of obstetrics and gynaecology Halland's hospital Halmstad).

**Contributors** VS, YC, LB, MZ, KL, AS, A-KW, HÖ, HF, OA, MV, MD, SBW and MB planned the study. YC, LB, MZ, KL and VS wrote the protocol. AS, A-KW, HÖ, HF, OA, MV, MD, SBW, MB and UÅ critically revised and accepted the final version for publication.

**Funding** The study has been financed by grants from the Swedish state under the agreement between the Swedish government and the county councils, the ALF agreement (YC, ALFGBG-77860; MZ, 2020-YF0016), Project Grant from the Department of Obstetrics and Gynaecology, Sahlgrenska Academy, Gothenburg University Sweden (VS) and regional research funding Western health care region (VS, VGFOUREG-938771). According to the Swedish Research Council decision spring 2020 regarding COVID-19 research; a defined funding was allowed from previous grants 2018-00470 (HF), 2016-00526 (SBW) and 2019-02082 (Simon Timpka).

**Disclaimer** The funders had no role in the study design; collection, management, analysis, and interpretation of data; writing of the report; and the decision to submit any manuscripts based on this study for publication.

**Competing interests** None declared.

**Patient consent for publication** Not required.

**Provenance and peer review** Not commissioned; externally peer reviewed.

**ORCID iDs**
Ylva Carlsson http://orcid.org/0000-0002-1414-7279
Lina Bergman http://orcid.org/0000-0001-5202-9428
Mehreen Zaigham http://orcid.org/0000-0003-0129-1578
Karolina Linden http://orcid.org/0000-0002-2792-3142
Ola Andersson http://orcid.org/0000-0002-3972-0457
Malin Veje http://orcid.org/0000-0001-5487-0616
Helena Fadl http://orcid.org/0000-0002-2691-7525
Magnus Domellöf http://orcid.org/0000-0002-0726-7029
Marie Blomberg http://orcid.org/0000-0003-4679-550X
Sophia Brismar Wendel http://orcid.org/0000-0002-9401-8062
Verena Sengpiel http://orcid.org/0000-0002-3608-7430

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
