## [Reviewer comments · BMJ Open]

ARTICLE DETAILS

TITLE (PROVISIONAL)	COVID-19 in Pregnancy and Early childhood (COPE) - study protocol for a prospective, multicentre biobank, survey and database cohort study
AUTHORS	Carlsson, Ylva; Bergman, Lina; Zaigham, Mehreen; Linden, Karolina; Andersson, Ola; Veje, Malin; Sandström, Anna; Wikström, Anna-Karin; Östling, Hanna; Fadl, Helena; Domellöf, Magnus; Blomberg, Marie; Brismar Wendel, Sophia; Åden, Ulrika; Sengpiel, Verena

VERSION 1 – REVIEW

REVIEWER	Wood, Rachael NHS National Services Scotland, Information Services Division
REVIEW RETURNED	17-Mar-2021

GENERAL COMMENTS	Thank you for the opportunity to review the protocol for this interesting study. It represents an important opportunity to get a wide range of complementary information (routine administrative data, biological sample data, and patient reported data) on a large population based sample of pregnant women with and without COVID-19 to address a range of linked research questions. I would make the following comments to help improve the protocol further – The process by which women will be identified for inclusion in the study (and then categorised as having/not having COVID) could be further clarified. In particular, will all women attending for antenatal or delivery care in a participating centre be invited to join the study each time they attend the hospital? If so, presumably the group of women ‘testing positive for SARS-CoV-2 during pregnancy’ (who are listed as separately identified for inclusion, specifically in the COVID group) are a subset of this group (assuming that the vast majority of pregnant women attend antenatal care at some time)? Do you have access to results of viral testing done in the participating maternity units only, or all testing regardless of location? More background information on how testing is provided in Sweden would be helpful. If it is all tests (e.g. through SmiNet), how do you know if a test was taken from a pregnant woman? Linked to the above, no dates for the study are provided. It would be helpful to know start and end dates for the recruitment period (recognising this may vary by centre). Again, linked to this it would be helpful to provide indicative information on how many women would therefore be expected to be invited to participate in the study. Do you have any preliminary
--

	information on the proportion invited that are agreeing to take part? Can any indication then be generated of the expected number of women with COVID who may be included in the study (based on known incidence in the general population)? Can you clarify to what extent women who lose their pregnancy prior to starting antenatal care (either through miscarriage or termination of pregnancy) will be included in your study? As you rightly note in the introduction and discussion sections, understanding of the impact of COVID in early pregnancy is a key knowledge gap at present. Linked to this, at what stage of pregnancy do women in Sweden typically engage with antenatal care (and hence have an opportunity to be included in the study)? In general, I missed having a clear statement of the outcomes to be assessed, and supporting definitions. I am aware that the potential list of outcomes to be considered may be very long, so it may not be feasible to provide a comprehensive list, but perhaps some key outcomes measures could be specified/defined. I was slightly confused about your planned duration of follow up. Detailed information is given for the 12 months following the end of pregnancy, but then some of the information you suggest linking to, and the indicative research questions included in the supplementary material, suggest much longer follow. It would be helpful to be clear what is within the current plans (and covered by current governance approvals) and what may be aspirations for the future. Finally, I found it difficult to reconcile the information provided in Table 2 and the text immediately following the table. The table is presented as providing details of the national registers/data sources that will be used to source data for linkage into the study dataset. The text below the table then lists these registers but also several other data sources. It would be helpful for Table 2 to cover all data sources and provide a clear rationale for what each will provide. This may link to the duration of follow up point above. I would emphasise however, that I think this is a very interesting proposal which will make an important contribution to our understanding of the impact of COVID in pregnancy.
--	--

REVIEWER	Gonçalves, Ana
REVIEW RETURNED	Universidade Federal do Rio Grande do Norte 22-Mar-2021

GENERAL COMMENTS	March 21, 2020. Dear Professor Editor We appreciate the opportunity to collaborate with this prestigious journal reviewing the Manuscript ID BMJ open- -2021-049376. entitled "COVID-19 in Pregnancy and Early childhood (COPE) - study protocol for a prospective multicentre biobank, survey, and database cohort study" The purpose of this project is to study the impact of COVID-19 on pregnancy outcomes and long-term maternal and child health by:  1) establishing a database and biobank from pregnant women with COVID-19 and presumably non-infected women and their infants. 2) studying how women and their partners experience pregnancy,
---

	childbirth, and early parenthood in the COVID-19-pandemic. After reading the article and evaluating the paper personally, we feel that the manuscript is a well-written article and needs minor revisions before it can be considered appropriate for potential publication. Best wishes Ana Katherine Gonçalves General aspects The protocol proposes an extremely pertinent idea and reasonable objectives. Very up-to-date and essential on the global stage. The latter study includes both parents in the population, valuing the couple's emotional aspect. The population included 98-100% of births in Sweden, ensuring the study's high fidelity to its internal reality. Great description of how and which samples will be taken. Specifics aspects: Page 2, L57 - Add keyword on Early childhood. Page 5, L15 - Only this limitation? Is it not possible to have a late refusal or loss of follow-up? Page 37, L34 - The questionnaire is brief but consistent. The study is a well-planned cohort proposition. However, we do not understand why it was registered as a trial since no intervention is being proposed. "Considering the NIH Definition of a Clinical Trial is a research study in which one or more human subjects are prospectively assigned to one or more interventions (which may include placebo or other control) to evaluate the effects of those interventions on health-related biomedical or behavioral outcomes." Please, clarify this point.
--	---

REVIEWER	Okunade, Kehinde University of Lagos, Department of Obstetrics and Gynaecology
REVIEW RETURNED	28-Mar-2021

GENERAL COMMENTS	 1. This manuscript and the study methodology should be written in a clearly understandable manner. 2. Is this a prospective or ongoing study? The use of tenses in the manuscript is confusing. 2. The study dates should be clearly stated. 3. Who are the proposed study participants? 4. How will the participants be selected into the study? 5. The process of obtaining participant consent is not clearly stated 6. The study outcomes should be explicitly defined 7. The manuscript will require further review by a native English speaker.
--

VERSION 1 – AUTHOR RESPONSE

Reviewer: 1

Dr. Rachael Wood, NHS National Services Scotland, University of Edinburgh

Thank you for reviewing our manuscript and recognising the importance of our study.

I would make the following comments to help improve the protocol further –

The process by which women will be identified for inclusion in the study (and then categorised as having/not having COVID) could be further clarified. In particular, will all women attending for antenatal or delivery care in a participating centre be invited to join the study each time they attend the hospital? If so, presumably the group of women ‘testing positive for SARS-CoV-2 during pregnancy’ (who are listed as separately identified for inclusion, specifically in the COVID group) are a subset of this group (assuming that the vast majority of pregnant women attend antenatal care at some time)? Do you have access to results of viral testing done in the participating maternity units only, or all testing regardless of location?

Thank you for pointing this out. We have re-written the Study design and Population section according to your suggestion, page 9ff.

We have added information on the recruitment process, test capacity in Sweden, and availability of test results at the antenatal care units. However, it was not possible for us to give more detailed data on “percentage of women approached” as this has varied across maternity units and from week to week secondary to the severity of the pandemic, resources at the hospitals actively participating in the study and regulations regarding research activity etc.

More background information on how testing is provided in Sweden would be helpful. If it is all tests (e.g. through SmiNet), how do you know if a test was taken from a pregnant woman?

We have tried to clarify this further at several different places of the revised manuscript.

To summarise, we will receive information on positive SARS-CoV-2 tests in pregnant women when: women are sampled at their antenatal care/maternity units as part of screening tests, self-testing since women are supposed to report all positive test results to their care providers (e.g their local midwife). The results from this information will define in which sampling group (COVID-19 or Screening) the woman will be included.

Similarly, at the time of statistical analyses, we will be able to divide women more accurately into a “COVID-19” and a non-infected “control group” after analysing the women’s antibody status at different time points during the pregnancy. We will also be able to import data from the mandatory SmiNet database using the woman’s Swedish personal identification number.

The manuscript has been revised as follows:

1) Regarding sampling strategies in Sweden, page 11: “Before June 2020, there was limited testing capacity in Sweden and only symptomatic patients admitted to the hospital were tested. Since June/July 2020, testing for SARS-CoV-2 has become widely available, even outside hospitals, to all citizens in Sweden. In the beginning of 2021, SARS-CoV-2 screening was introduced for all patients admitted to Swedish hospitals including pregnant women upon admission to maternity units. All adults, including pregnant women, are currently required to take a SARS-CoV-2 test in case of symptoms. Detailed data on the number of performed tests and test-positivity in different regions, age groups and over time are available at the homepage of the Public Health Agency of Sweden (30).”

2) When defining the “COVID-19 group”, page 11: “This group will include women that 1) test positive for SARS-CoV-2 during pregnancy or at delivery, 2) have a positive SARS-CoV-2 antibody test from infection during the current pregnancy, 3) have COVID-19 as a “clinical diagnose” at the

time point of delivery before test results are available.”

3) When defining the “Screening group”, page 11f: “This group consists of women without symptoms and/or with a negative test for SARS-CoV-2 during the current pregnancy. These women are recruited at participating centres during their antenatal care check-up or during their visit to the maternity unit. A woman in the Screening group may be included into the COVID-19 group later on during the pregnancy if she contracts COVID-19. This may also be the case at the time of statistical analyses, in case the biobank specimens should indicate an asymptomatic SARS-CoV-2 infection or a positive test result is found registered in the Swedish Register for mandatory registration of notifiable infectious diseases (SmiNet).”

4) Under “Data processing and analysis”, “Exposure definition in regard to clinical outcomes”, page 24: “Based on RT-PCR and serology and data from SmiNet, women will be divided into infected women (COVID-19 during pregnancy) and non-infected women (no COVID-19 during pregnancy).”

Linked to the above, no dates for the study are provided. It would be helpful to know start and end dates for the recruitment period (recognising this may vary by centre).

We agree that this is important information and have added the following text under “Study design and population”, page 9: “Patient recruitment formally started on June 1st, 2020.”

Page 10: “The primary goal is to recruit 200 women in the “COVID-19 group” and 1000 women in the “Screening group”. Further recruitment to the biobank part will depend on adequate funding, study centre capacity and the overall progress of the pandemic. Inclusion to the questionnaire part of the study will continue until the obstetric and neonatal departments return to their pre-pandemic routines and social restrictions due to the COVID-19 pandemic are revoked.”

Again, linked to this it would be helpful to provide indicative information on how many women would therefore be expected to be invited to participate in the study. Do you have any preliminary information on the proportion invited that are agreeing to take part? Can any indication then be generated of the expected number of women with COVID who may be included in the study (based on known incidence in the general population)?

As described above, we will not be able to provide the exact number of women approached for study participation. Recruitment possibilities change on a daily basis depending on the situation of the pandemic and whether the hospitals are able to provide research staff. Further, a considerable number of study participants contacted study centres actively after learning about the study in social/mass media without having been individually approached.

However, until May 10th 2021, 1621 women had been recruited to the questionnaire part (of these n=570 in the COVID-19 group and n=1051 in the Screening group) and 1408 women are currently participating in the biobank part (n=524 in the COVID-19 group and n=884 in the Screening group). With respect to the partners, currently 626 have participated in the questionnaire part, n=242 in the COVID-19 group and n=384 in the Screening group.

We have not started analysing the biosamples yet, so we do not know the exact percentage of women with asymptomatic SARS-CoV-2 infection in the Screening group.

Can you clarify to what extent women who lose their pregnancy prior to starting antenatal care (either through miscarriage or termination of pregnancy) will be included in your study? As you rightly note in the introduction and discussion sections, understanding of the impact of COVID in early pregnancy

is a key knowledge gap at present. Linked to this, at what stage of pregnancy do women in Sweden typically engage with antenatal care (and hence have an opportunity to be included in the study)?

According to the Swedish Pregnancy Register, the mean gestational age at the time of the first antenatal care visit is 8.8 weeks. All pregnant women, independent of week of gestation, are eligible for inclusion into the COPE study. Women are approached regarding inclusion at the time we get knowledge of a positive SARS-CoV-2 test – independent of gestational age. Active recruitment to the Screening group is organised differently at different study sites. At many centres, recruitment can occur in connection to the 1st and/or 2nd trimester routine ultrasound. However, as stated above, many women have actively contacted us for inclusion - at different weeks of gestation, since the study has caught the interest of the media here in Sweden.

As described in the manuscript, samples taken in clinical routine in the beginning of antenatal care might be retrieved for analyses in the COPE study so that we will have biological samples for early pregnancy even in case a woman enters the study during the second or third trimester.

Miscarriage or abortion are no reason for exclusion; however, the majority of women is included after the first trimester.

In general, I missed having a clear statement of the outcomes to be assessed, and supporting definitions. I am aware that the potential list of outcomes to be considered may be very long, so it may not be feasible to provide a comprehensive list, but perhaps some key outcomes measures could be specified/defined.

We notice that the reference to the supplemental material 3 was missing in the manuscript. We have now added a section "Research questions" under "Data processing and analysis" to answer your suggestion, page 23: "Research questions The COPE study is collecting information for a database and biobank in order to study the association of COVID-19 during pregnancy with a wide variety of different pregnancy, maternal and neonatal outcomes including the long-term follow-up of maternal and child health, as well as parental experience. Predefined research questions concern the incidence of infection and COVID-19 at different time points of the pandemic; the impact of COVID-19, gestational age at infection, severity of disease, viral load, presence of SARS-CoV-2 and/or antibodies against SARS-CoV-2 in different compartments of the mother and/or child, pregnancy outcome, maternal and child health; experience of childbirth and early parenthood during the pandemic, see Supp material 3 for details.

www.sahlgrenska.se Sahlgrenska Universitetssjukhuset Område 1, VO Obstetrik ADRESS Diagnosvägen 15, 416 85 Göteborg TELEFON växel 031-342 10 00 Exposure definition in regard to clinical outcomes Based on RT-PCR and serology and data from SmiNet, women will be divided into infected women (COVID-19 during pregnancy) and non-infected women (no COVID-19 during pregnancy). In some analyses/subanalyses, gestational age at infection, severity of disease, viral load and immune response will be considered as an additional exposure variables for the infected group.

Examples of outcome definition

- Pregnancy and neonatal outcomes will be retrieved from the SPR and SNQ, either registered as tick-boxes, actual measures or ICD10 codes: Preeclampsia (ICD10 O14), gestational age at birth, preterm delivery (with subanalyses for early, moderate, late preterm delivery as well as spontaneous vs iatrogenic preterm delivery), birthweight, small for gestational age, birth asphyxia, and perinatal death.

- Thromboembolic event diagnosis will be retrieved from the SPR and the National Patient Register (ICD10 I82, I26).

- Vertical transmission as defined by Shah et al. (38)

- Child development and health: Developmental delays or potential delays as measured by ASQ up to four years of age. Neurological disorders diagnosed during the first four years of life (composite) retrieved from the National Patient Register; Any mental or behavioural disorder (ICD10 F00-F99), impaired vision (H54), impaired hearing (H90, H91), cerebral palsy and other paralytic syndromes (G80-G83).
- Presence of antibodies in umbilical cord blood (IgG, IgM) and breast milk (IgG, IgM, IgA) Subanalyses will be performed to study the impact of country of birth, socioeconomic status or underlying disease in the mother (for example obesity, hypertension, diabetes, asthma)."

I was slightly confused about your planned duration of follow up. Detailed information is given for the 12 months following the end of pregnancy, but then some of the information you suggest linking to, and the indicative research questions included in the supplementary material, suggest much longer follow. It would be helpful to be clear what is within the current plans (and covered by current governance approvals) and what may be aspirations for the future.

Thank you for pointing out the need for clarification regarding this topic.

Since we submitted the manuscript, we got approval from the Swedish Ethical Review Authority to extend the questionnaire follow-up for parents and children until four years after delivery. This information was added under the Questionnaires section, page 19, and in table 3, page 22: "

Questionnaires

Upon inclusion, women and their partners from both the COVID-19 group and the Screening group, are asked to fill out different electronic questionnaires up to four years after delivery."

We have added the following description of the study design, page 9: "Data are collected in four different ways: 1) biosampling, 2) survey-based follow-up until four years after delivery, 3) linkage to Swedish health and quality registers enabling long-term follow-up, and 4) interviews." We added clarification regarding long-term follow-up under "Register- and medical record data on obstetric, medical, and neonatal outcome"s section, page 15: "Biobank laboratory analyses and questionnaire results will be linked to register data using Swedish personal identification numbers in order to follow long-term maternal and child health as well as child growth and development." All research questions specified in supplemental material 3 have been approved by the Ethics Review Board.

Finally, I found it difficult to reconcile the information provided in Table 2 and the text immediately following the table. The table is presented as providing details of the national registers/data sources that will be used to source data for linkage into the study dataset. The text below the table then lists these registers but also several other data sources. It would be helpful for Table 2 to cover all data sources and provide a clear rationale for what each will provide. This may link to the duration of follow up point above.

We added information on all registers as well as examples for variables that will be retrieved from the different registers to Table 2, page 16ff. Due to the number of research questions, it is not possible to describe all variables in detail within the scope of this manuscript. A comprehensive description of this type of data will have to be part of the different manuscripts presenting results from the COPE study.

I would emphasise however, that I think this is a very interesting proposal which will make an important contribution to our understanding of the impact of COVID in pregnancy.

Thank you for taking the time to review our manuscript.

Reviewer: 2 Dr. Ana Gonçalves, Universidade Federal do Rio Grande do Norte

Thank you for reviewing our manuscript and recognising the importance of our study.

General aspects

The protocol proposes an extremely pertinent idea and reasonable objectives. Very up-to-date and essential on the global stage. The latter study includes both parents in the population, valuing the couple's emotional aspect.

The population included 98-100% of births in Sweden, ensuring the study's high fidelity to its internal reality. Great description of how and which samples will be taken.

Specifics aspects:

Page 2, L57 - Add keyword on Early childhood. www.sahlgrenska.se Sahlgrenska Universitetssjukhuset Område 1, VO Obstetrik ADRESS Diagnosvägen 15, 416 85 Göteborg TELEFON växel 031-342 10 00 We added "childhood" to the keywords as we will follow the children throughout childhood based on the Swedish quality and health registers.

Page 5, L15 - Only this limitation? Is it not possible to have a late refusal or loss of follow-up?

We have added the following text regarding the methodological strengths and limitations of the biological sampling and questionnaire-based follow-up included in the study, page 4-5:

- Logistics provided by Hospital Integrated Biobank Sweden enable high quality biological sampling at several time points during pregnancy according to standardised protocols. However, due to resource limitations at the hospitals during the pandemic, some women will not have complete samples from all time points of interest.
- Based on validated instruments, child health and development during the first four years of life will be reported by parents along with comprehensive register-based long-term follow-up. There is a risk of selection bias regarding the follow-up questionnaires where we expect that a proportion of the study population will not answer the questionnaires. Regarding the questionnaire part there definitely will be loss to follow-up. However, other outcomes will be collected based on mandatory health registers, e.g., in case the family does not leave the study or move to another country, there will be no loss to follow-up.

Page 37, L34 - The questionnaire is brief but consistent.

The study is a well-planned cohort proposition. However, we do not understand why it was registered as a trial since no intervention is being proposed. "Considering the NIH Definition of a Clinical Trial is a research study in which one or more human subjects are prospectively assigned to one or more interventions (which may include placebo or other control) to evaluate the effects of those interventions on health-related biomedical or behavioral outcomes." Please, clarify this point.

As indicated on clinicaltrials.gov this study is an observational – as opposed to interventional – study. <https://prsinfo.clinicaltrials.gov/definitions.html#StudyType> We decided to register our study as we wanted to inform other researchers on this ongoing project as soon as possible and thus enable future collaboration. We have already been contacted by researchers who found our study at the Clinical Trials page.

Reviewer: 3 Dr. Kehinde Okunade, University of Lagos

1. This manuscript and the study methodology should be written in a clearly understandable manner. We tried to clarify several aspects of the methodology and analyses planned within the COPE study. Please see changes marked in red in the “Methods and Analysis” section of the manuscript, page 9ff.

2. Is this a prospective or ongoing study? The use of tenses in the manuscript is confusing. Thank you for pointing out the need for clarification.

COPE is an ongoing study as we now have specified on page 9: “The COPE study is an ongoing Swedish multicentre study, facilitated by the Swedish network for national clinical studies in Obstetrics and Gynaecology (SNAKS, www.snaks.se).”

Further, we added information on study start and planned time for recruitment, page 9: “Patient recruitment formally started on June 1st, 2020.”

Page 10: “The primary goal is to recruit 200 women in the “COVID-19 group” and 1000 women in the “Screening group”. Further recruitment to the biobank part will depend on adequate funding, study centre capacity and the overall progress of the pandemic. Inclusion to the questionnaire part of the study will continue until the obstetric and neonatal departments return to their pre-pandemic routines and social restrictions due to the COVID-19 pandemic are revoked.”

Currently, we are at the stage of data-collection. We changed to present tense throughout the part regarding recruitment and biosampling and kept future tense when referring to analyses of biological samples and statistical analyses which will take place in the future.

2. The study dates should be clearly stated.

Please see the answer above.

3. Who are the proposed study participants?

We are sorry if this was not entirely clear. We elaborated the method section on recruitment, please see page 10:

“All women, aged 18 years or older, receiving antenatal care or giving birth at participating centres are eligible for the study.” Page 10: “Partners aged 18 years or older, are also eligible for participation.”

4. How will the participants be selected into the study?

Thank you for pointing this out as unclear. We now provide a more detailed description of the recruitment procedure, please see page 11:

“The COVID-19 group: This group will include women that 1) test-positive for SARS-CoV-2 during pregnancy or at delivery, 2) have a positive SARS-CoV-2 antibody test from infection during the current pregnancy, 3) have COVID-19 as a “clinical diagnose” at the time point of delivery before test results are available. Before June 2020, there was limited testing capacity in Sweden and only symptomatic patients admitted to the hospital were tested. Since June-July 2020, testing for SARS-CoV-2 has become widely available, even outside hospitals, to all citizens in Sweden. In the beginning of 2021, SARSCoV-2 screening was introduced for all patients admitted to Swedish hospitals including pregnant women upon admission to maternity units. All adults, including pregnant women, are currently required to take a SARS-CoV-2 test in case of symptoms. Detailed data on the number of performed tests and test-positivity in different regions, age groups and over time are available at the homepage of the Public Health Agency of Sweden (30). Due to restrictions on research-related appointments during the pandemic, women are recruited to the COVID-19 group when they seek inpatient care or in connection with their routine antenatal care visit. In the later case, they are included when they are in remission from COVID-19.

The Screening group:

This group consists of women without symptoms and/or with a negative test for SARS-CoV-2 during the current pregnancy. These women are recruited at participating centres during their antenatal care check-up or during their visit to the maternity unit.

A woman in the Screening group may be included into the COVID-19 group later on during the pregnancy if she contracts COVID-19. This may also be the case at the time of statistical analyses, in case the biobank specimens should indicate an asymptomatic SARS-CoV-2 infection or a positive test result is found registered in the Swedish Register for mandatory registration of notifiable infectious diseases (SmiNet).”

5. The process of obtaining participant consent is not clearly stated

We tried to clarify the process of obtaining consent in the “Study design and population” section, please see page 10:

“Participants have access to study information which is freely available in the waiting rooms of the antenatal care and maternity units involved in the study, the COPE study homepage (www.copestudien.se), social media, interviews and articles available in mass media along with active recruitment by the local study research team. www.sahlgrenska.se Sahlgrenska Universitetssjukhuset Område 1, VO Obstetrik ADRESS Diagnosvägen 15, 416 85 Göteborg TELEFON växel 031-342 10 00

Recruitment of pregnant women may occur at different time points during the pregnancy, e.g. during the first or second trimester ultrasound screening visit, upon admission for pregnancy complications or admission to the delivery/COVID-19 units of any of the participating hospitals. Partners aged 18 years or older, are also eligible for participation. Participating women and their partners receive oral and written information about the study and are required to provide written consent. Women can choose to participate in either the biobank or the questionnaire part of the study, or both.”

6. The study outcomes should be explicitly defined

Thank you for pointing this out. Please see even answer to reviewer 1.

The COPE study collects data to a biobank and database, rather than for a specific research question. Thus, the research questions are not limited to the questions specified in the manuscript. The general aim is to study the impact of SARS-CoV-2 infection and COVID-19 on pregnancy outcomes and long-term maternal and child health as well as studying how women and their partners experience pregnancy, childbirth and early parenthood in the COVID-19-pandemic.

A number of pre-specified research questions is provided in supplemental material 3. Due to the number of research questions, it is not possible to describe all variables in detail within the scope of this manuscript. A comprehensive description of this type of data will have to be part of the manuscript presenting results from the COPE study.

However, we added examples of variables and outcomes to table 2 and added a section on some of the main outcomes of interest to the “Data processing and analysis section”, please see page 23: “Research questions The COPE study is collecting information for a database and biobank in order to study the association of COVID-19 during pregnancy with a wide variety of different pregnancy, maternal and neonatal outcomes including the long-term follow-up of maternal and child health, as well as parental experience.

Predefined research questions concern the incidence of infection and COVID-19 at different time points of the pandemic; the impact of COVID-19, gestational age at infection, severity of disease, viral load, presence of SARS-CoV-2 and/or antibodies against SARS-CoV-2 in different compartments of the mother

and/or child, pregnancy outcome, maternal and child health; experience of childbirth and early parenthood during the pandemic, see Supp material 3 for details.

Exposure definition in regard to clinical outcomes

Based on RT-PCR and serology and data from SmiNet, women will be divided into infected women (COVID-19 during pregnancy) and non-infected women (no COVID-19 during pregnancy). In some analyses/subanalyses, gestational age at infection, severity of disease, viral load and immune response will be considered as an additional exposure variables for the infected group. Examples of outcome definition

- Pregnancy and neonatal outcomes will be retrieved from the SPR and SNQ, either registered as tick-boxes, actual measures or ICD10 codes: Preeclampsia (ICD10 O14), gestational age at birth, preterm delivery (with subanalyses for early, moderate, late preterm delivery as well as spontaneous vs iatrogenic preterm delivery), birthweight, small for gestational age, birth asphyxia, and perinatal death.
- Thromboembolic event diagnosis will be retrieved from the SPR and the National Patient Register (ICD10 I82, I26).
- Vertical transmission as defined by Shah et al. (38)
- Child development and health: Developmental delays or potential delays as measured by ASQ up to four years of age. Neurological disorders diagnosed during the first four years of life (composite) retrieved from the National Patient Register; Any mental or behavioural disorder (ICD10 F00-F99), impaired vision (H54), impaired hearing (H90, H91), cerebral palsy and other paralytic syndromes (G80-G83).
- Presence of antibodies in umbilical cord blood (IgG, IgM) and breast milk (IgG, IgM, IgA) Subanalyses will be performed to study the impact of country of birth, socioeconomic status or underlying disease in the mother (for example obesity, hypertension, diabetes, asthma)."

7. The manuscript will require further review by a native English speaker.

A native English speaker within the COPE study group has revised the manuscript.

VERSION 2 – REVIEW

REVIEWER	Wood, Rachael NHS National Services Scotland, Information Services Division
REVIEW RETURNED	11-Aug-2021
GENERAL COMMENTS	The authors have provided a comprehensive response to reviewer comments, and the paper is now much clearer. I recommend publication, with no further amendments required.